# Preparation of a Biomedical Scaffold from High-Molecular-Weight Poly-DL-Lactic Acid Synthesized via Ring-Opening Polymerization

**DOI:** 10.3390/polym17121708

**Published:** 2025-06-19

**Authors:** Geraldine Denise Bazan-Panana, Manuel J. Torres-Calla, María Verónica Carranza-Oropeza

**Affiliations:** 1Department of Chemical Engineering, Major National University of San Marcos, Lima 15081, Peru; mcarranzao@unmsm.edu.pe; 2Faculty of Physical Sciences, Major National University of San Marcos, Lima 15081, Peru; manuel.torres@unmsm.edu.pe; 3Centro de Investigaciones Tecnológicas, Biomédicas y Medioambientales, Callao 07006, Peru

**Keywords:** poly-DL-lactic acid, ring-opening polymerization, scaffold, tissue engineering, 3D printing

## Abstract

In this study, poly-DL-lactic acid (PDLLA) was synthesized via ring-opening polymerization (ROP) to develop a biomedical scaffold for tissue engineering. A rotary evaporator with a two-stage vacuum pump under an inert atmosphere and constant stirring was used. A factorial design with three factors (oligomerization time, ROP time, and catalyst concentration) at two levels was applied. Polymers were characterized by FTIR, capillary viscometry, ^1^H-NMR, DSC, and TGA. The kinetic study revealed a first-order model, indicating that the polymerization rate depends linearly on monomer concentration. The activation energy (70.5 kJ/mol) suggests a moderate energy requirement, consistent with ring-opening polymerization, while the high pre-exponential factor (6.93 × 10^6^ min^−1^) reflects a significant frequency of molecular collisions. The scaffold was fabricated via extrusion and 3D printing, and its morphology, porosity, mechanical properties, and contact angle were studied. The highest molecular weight PDLLA was obtained with 6 h of oligomerization, 4 h of ROP, and 1% catalyst concentration. The samples exhibited thermal stability below 40 °C, while the scaffold reached 71.6% porosity, an 85.97° contact angle, and a compressive strength of 4.24 MPa with an elastic modulus of 51.7 MPa. These findings demonstrate the scaffold’s potential for biomedical applications.

## 1. Introduction

Bone regeneration in cases of osteoporosis and irreversible bone defects represents one of the greatest challenges in tissue engineering [1,2,3]. Currently, metallic implants are the most widely used solution; however, they present significant limitations such as the risk of rejection, the need for additional surgeries, and poor integration with natural bone [4,5]. Meanwhile, within the framework of tissue engineering, scaffolds have been shown to guide the development of new bone tissue by providing mechanical stability and a three-dimensional environment for cells, enabling them to adhere, proliferate, differentiate, and secrete extracellular matrix, ultimately leading to tissue regeneration [6]. In this context, scaffolds made from biodegradable biomaterials such as polylactic acid (PLA) have emerged as a promising alternative due to their biocompatibility and ability to degrade into non-toxic products within the body. Approved by the FDA, PLA has proven to be an efficient material for biomedical applications and has been extensively studied for the fabrication of scaffolds for tissue regeneration [7,8,9].

PLA can be synthesized through various methods, including condensation polymerization, azeotropic dehydrative condensation, and ring-opening polymerization (ROP) [10]. While the first methods present limitations due to the presence of toxic impurities and the low molecular weight of the resulting polymer, ROP has proven to be the most efficient method for obtaining high-molecular-weight PLA, particularly when using catalysts such as tin(II) 2-ethylhexanoate, which is FDA-approved [7,8,11].

The objective of this study is to synthesize high-molecular-weight poly-DL-lactic acid (PDLLA) via ring-opening polymerization (ROP) and to explore the influence of key reaction parameters, including oligomerization conditions, reaction times, and catalyst concentration [12,13,14,15,16,17,18,19,20]. Furthermore, the study aims to fabricate and characterize 3D-printed PDLLA scaffolds, evaluating their morphological and mechanical properties in comparison to human trabecular bone [21,22].

This research is relevant because it provides a detailed analysis of the synthesis–process–structure relationship in PDLLA-based scaffolds, which are intended for bone regeneration applications. The development of customized, biodegradable scaffolds offers a potential alternative to conventional metallic implants, particularly for patients with osteoporosis, thereby contributing to safer and more effective regenerative strategies in tissue engineering [23,24].

## 2. Materials and Methods

### 2.1. Experimental Design

The statistical software Minitab 19 was used to design the experiments and perform the statistical analysis of the synthesis results. The aim of the experimental design was to identify the key parameters influencing the synthesis of PDLLA. The most relevant variables were selected based on preliminary tests, which helped to observe factor trends and determine those with the greatest impact. A full factorial design was implemented to explore the conditions under which the polymer with the highest molecular weight was obtained.

### 2.2. Reagents

The DL-lactic acid was supplied by Sigma-Aldrich (St. Louis, MO, USA). As for catalysts, tin(II) chloride dihydrate (98% purity) was purchased from HIMEDIA (Kelton, PA, USA), and stannous octoate from Sigma-Aldrich was imported from São Paulo, Brazil. The ring-opening initiator, 1-octanol, was acquired from Sigma-Aldrich.

### 2.3. Synthesis of Poly(DL-lactide) by Ring-Opening Polymerization

The synthesis was carried out in three stages: polycondensation, depolymerization or thermal cracking, and ring-opening polymerization (ROP). During the synthesis, the working pressure was 15 μmHg (microns of mercury), and continuous stirring was maintained at 100 rpm. A DLAB rotary evaporator was utilized, with a temperature range from room temperature to 180 °C, offering a temperature control precision for oil of ±3 °C and a rotation speed range from 20 to 200 rpm. This equipment was coupled with a two-stage pump (model VE245N, ½ HP) from VALUE Industrial Co., Ltd. (Zhejiang, China). Additionally, an inert atmosphere of N_2_ was essential to provide an oxygen- and moisture-free environment to prevent oxidation or decomposition of the product.

#### 2.3.1. Polycondensation of Lactic Acid

An amount of 50 mL of DL-lactic acid was added to a round bottom flask. The dehydration reaction was carried out to remove the water content of the monomer. An initial temperature of 80 °C was chosen, and this was gradually and carefully increased to 120 °C, without exceeding the boiling temperature of lactic acid (122 °C). Subsequently, the temperature was increased to 160 °C to form oligomers, with reaction times of 4 and 6 h. The reaction product was dissolved with 25 mL of acetone and precipitated in 150 mL of distilled water with vigorous stirring at 700 rpm. Then, the precipitate was recovered by vacuum filtration using Whatman 42 filter paper and stored in a desiccator for 24 h to eliminate moisture.

#### 2.3.2. Depolimerization of the PDLLA Oligomer

The obtained oligomer from Stage 1 was heated in a round-bottom flask at a temperature of 180 °C for 7 h, with the addition of SnCl_2_∙2H_2_O 1% *w*/*w* as a catalyst. Then, lactide was purified by recrystallization using 30 mL of ethyl acetate at its boiling point (77.2 °C) as a solvent. After cooling the solution, the crystals were isolated by vacuum filtration and dried at 40 °C for 24 h.

#### 2.3.3. Polymerization by the Ring-Opening Method of Lactide (ROP)

High-molecular-weight PDLLA was obtained via ROP, using stannous octoate catalyst (1% and 2% *w*/*w*) and 1-octanol (1% *w*/*w*) as initiator. The lactide obtained in Stage 2, the catalyst, and the initiator were added to a round-bottom glass flask with a reaction temperature of 140 °C for 4 and 6 h. After completing the synthesis, the PDLLA acid obtained was dissolved in 25 mL of chloroform, precipitated in 80 mL of methanol, separated by vacuum filtration, and finally, dried in an oven at 60 °C for 12 h.

### 2.4. Characterization of PDLLA

#### 2.4.1. Fourier-Transform Infrared Spectroscopy (FTIR)

To confirm that the produced material was PDLLA, the FTIR technique was used, specifically the OPUS software version 7.5 of the Bruker ALPHA II compact FT-IR spectrometer (Bruker Optik GmbH, Ettlingen, Germany). The characteristic peaks expected in our samples include −OH at 3500 cm^−1^; asymmetric and symmetric −CH stretching of the alkane group at 3000 and 2950 cm^−1^, respectively; the ester carbonyl group (C=O) at 1750 cm^−1^; asymmetric and symmetric −CH_3_ of the alkane group at 1450 cm^−1^ and 1350 cm^−1^, respectively; and the asymmetric and symmetric ester alkoxy group (C−O−C) at 1190 and 1090 cm^−1^, respectively.

#### 2.4.2. Capillary Viscometry

An Ubbelohde viscometer (SI-Analysis Typ 530 10/I, SI Analytics, Mainz, Germany) was used, which was placed in a temperature bath at 25 °C and secured with a universal stand. For each sample, four polymer solutions were prepared at different concentrations (0.3%, 0.5%, 0.7%, and 1%), using high-purity chloroform as the solvent. The flow times of the pure solvent and polymer solutions were then measured using a stopwatch. Equation (1) was applied to calculate the reduced viscosity (η_red_) as a function of polymer solution concentration (C), followed by a graphical determination of the intrinsic viscosity:(1)$$\eta_{red}=\frac{t_{1}-t_{0}}{t_{0}\cdot C}$$
where t_1_ is the time it takes for the solution to pass through a fixed volume of the capillary viscometer, and t_0_ is the time required for the pure solvent to pass through the same volume. Using Equation (2), with K = 0.0066 and α = 0.67, the average molecular weight (M) of each sample was calculated. These Mark–Houwink parameters were selected based on data reported in the *Polymer Data Handbook* [25], which compiles experimental values obtained for atactic polylactide in chloroform at 25 °C. The use of atactic parameters is appropriate in this case, as the synthesized PDLLA is an amorphous polymer composed of a random sequence of D- and L-lactic acid units, resulting in a microstructure with no stereoregular order. This random configuration corresponds to an atactic polymer, for which the selected K and α values are applicable [26].(2)$$M=\sqrt[{\alpha }]{\frac{\left[\eta \right]}{K}},$$

#### 2.4.3. Proton Nuclear Magnetic Resonance Spectroscopy (^1^H-NMR)

To obtain structural information on the synthesized polymer, ^1^H-NMR spectroscopy was performed. A total of 82 mg of PDLLA sample was weighed and placed in a vacuum desiccator equipped with a temperature controller. The sample was dried for 24 h at 45 °C. Subsequently, 12 mg of the dried polymer was dissolved in 600 μL of anhydrous chloroform (CDCl_3_), corresponding to a concentration of 0.02 mg/μL. Spectra were acquired at 25 °C using a Bruker Avance III HD 500 MHz spectrometer equipped with a helium-cooled TCI cryoprobe (Bruker BioSpin GmbH, Rheinstetten, Germany).

#### 2.4.4. Differential Scanning Calorimetry (DSC)

Approximately 5 mg of the samples was placed in aluminum pans and subjected to a heating scan from 20 to 250 °C at a rate of 5 °C/min. In each of the 12 DSC thermograms, the glass transition temperature (Tg) of PLA should be between 49 and 56 °C and is represented by the first peak. The melting temperature (Tm) and the enthalpy of fusion (ΔHm) are represented by the second peak, which, for the polymer studied, is observed between 132 and 146 °C. Additionally, the degree of crystallinity (χ) was calculated from the ΔHm values using Equation (3):(3)$$\chi =\frac{{\Delta H}_{m}}{{{\Delta H}_{m}}^{{}^{\circ}}}\times 100\%,$$
where ΔHm° is the enthalpy of fusion of 100% crystalline PLA (93.7 J/g).

#### 2.4.5. Thermogravimetric Analysis (TGA)

This technique was used to determine the thermal stability of PDLLA within a specific temperature range, as well as its degradation temperature. Since the resulting polymer is intended for the fabrication of a biomedical scaffold, it is essential that it does not undergo significant degradation at temperatures near body temperature, approximately 37 °C. For each scan, an initial sample mass of 5 mg was used, and the samples were subjected to a heating scan from 20 to 600 °C at a rate of 10 °C/min.

### 2.5. Kinetic Study

The synthesis of PDLLA was conducted at four reaction temperatures during the ROP stage (150, 160, 170, and 180 °C). Using a syringe, samples were taken every 20 min to monitor the reaction. The molecular weight of each sample was determined by capillary viscometry, and the degree of polymerization was calculated to establish the reaction order based on the integrated rate law. Subsequently, the reaction rate constants were calculated for each temperature according to the determined reaction order. The reaction rate was then plotted as a function of time to assess its influence on the polymerization kinetics. Finally, the linearized Arrhenius equation was applied to determine the Arrhenius constants (Ea and A) by plotting ln(kp) vs. 1/T.

### 2.6. Fabrication of a Polymeric Scaffold

The poly-DL-lactic acid (PDLLA) samples obtained under the selected synthesis conditions were crystallized at 130 °C for 30 min. Subsequently, the product was fed into an extruder at a temperature of 195 °C, with an extrusion speed of 15 rpm, resulting in the production of PDLLA filaments.

The 3D printing process was carried out using a model designed in OpenSCAD, version 2021.01, where a low-height cylindrical scaffold with a defined pore structure was created. The filaments obtained in Stage 4 were loaded into the 3D printer and used to fabricate the scaffolds by fused deposition modeling (FDM). This technique, increasingly applied in tissue engineering, enables the production of porous structures with controlled geometry and interconnectivity [27].

### 2.7. Characterization of Scaffolds

#### 2.7.1. Scanning Electron Microscopy (SEM)

The Prisma E scanning electron microscope from Thermo Fisher Scientific (Waltham, MA, USA), specifically designed for materials science applications, was used. After placing the sample in the sample holder, it was introduced into the chamber. SEM images were captured at a working distance of approximately 10 mm, with magnifications of 40× and 70×, providing an overview of the scaffold structure as well as insights into pore distribution and surface morphology. The equipment operated at a voltage of 20 kV and a pressure of 0.10 mbar, indicating that the images were acquired under low vacuum conditions. Additionally, a spot size of 5 was used, referring to the electron beam diameter on the sample, representing an optimal balance between signal and resolution.

#### 2.7.2. Porosity

The porosity of the scaffold was measured using the indirect method based on Archimedes’ principle, following the standard procedure described in [28]. Archimedes’ principle states that a body fully or partially submerged in a fluid experiences an upward force (buoyancy) equal to the weight of the displaced fluid.

#### 2.7.3. Mechanical Properties

Since bone repair scaffolds are used as implants, their mechanical response to compression is of critical importance. The tests were conducted following the procedure described in [29]. A ZwickRoell universal testing machine, model Z050 (ZwickRoell GmbH & Co. KG, Ulm, Germany), was used. The tests were performed with a 2.5 kN load cell at a speed of 1 mm/min. Measurements were carried out under controlled environmental conditions, with a temperature of 22 °C and a relative humidity of 50%.

#### 2.7.4. Surface Wettability

To determine the water contact angle, a drop of deionized water (5 µL) was deposited on the surface of the scaffolds at three different positions using a micropipette. Subsequently, the ImageJ software version 1.53t was used for analysis [23].

## 3. Results and Discussion

### 3.1. Experimental Design

A full factorial experimental design was developed considering three factors: oligomerization time, ROP polymerization time, and tin octoate catalyst percentage, each with two levels. Table 1 presents the quantities used in the polymer synthesis process, along with their coded representations, which serve as a reference for subsequent characterizations.

### 3.2. Synthesis of Poly(DL-lactide) by Ring-Opening Polymerization

At the end of the polycondensation process, both products exhibited a viscous consistency, which became more pronounced upon cooling, as shown in Figure 1a. To facilitate further processing, the product was dissolved in acetone (Figure 1b), forming a highly adhesive gel that fully dissolved within minutes. The resulting solution was then poured into distilled water under continuous stirring at 1000 rpm, gradually decreasing the speed. When using the oligomer obtained after 6 h of reaction, a compact, cloud-like precipitate was successfully formed (Figure 1c). However, the product synthesized after only 4 h did not achieve the same consistency. These findings indicate that the stirring speed plays a crucial role in obtaining the desired precipitation, with an initial agitation of 1000 rpm followed by a gradual reduction being optimal. Vacuum filtration was performed without complications for both oligomers (4 h and 6 h reactions). However, the precipitate from the 4 h reaction was loosely structured, whereas the 6 h sample formed a more compact solid (Figure 1d). The filtered precipitate was then transferred from the filter paper to a Petri dish (Figure 1e) and placed in a desiccator for 24 h to eliminate residual moisture. Finally, the dried oligomer (Figure 1f) was weighed to establish a correlation between its final mass and the initial mass of lactic acid used in the synthesis.

During the depolymerization stage, the oligomer obtained in the previous step (Figure 2a) was heated to 180 °C along with the catalyst, tin(II) chloride dihydrate, leading to the formation of the dimer, lactide (Figure 2b). This intermediate compound is crucial for the subsequent stage, where chain growth occurs in a controlled and homogeneous manner, resulting in a polymer with a high molecular weight. After synthesis, the product was dissolved in boiling ethyl acetate (Figure 2c). As the temperature gradually decreased, crystalline structures began to form (Figure 2d), which were then refrigerated to accelerate recrystallization (Figure 2e). The final step involved vacuum filtration, followed by 24 h of drying in a desiccator to obtain the purified, dry lactide (Figure 2f). Significant differences were observed between the products obtained when using the 4 h and 6 h oligomers. The 4 h oligomer failed to form long chains, leading to premature chain breakage and degradation during the 7 h depolymerization process at 180 °C. Although the same post-treatment procedures were applied, the amount of lactide recovered from the 4 h reaction was notably lower compared to that obtained from the 6 h oligomer.

The ring-opening polymerization (ROP) of lactide was carried out using the purified lactide obtained in the previous stage (Figure 3a). Two different concentrations of the catalyst, tin(II) octoate (Figure 3b), were used along with the initiator, 1-octanol (Figure 3c), to facilitate polymerization. After completion of the reaction, the resulting polymer was dissolved in chloroform (Figure 3d) and subsequently precipitated in methanol under vigorous stirring (Figure 3e). The precipitate was recovered through vacuum filtration, yielding a fine white powder (Figure 3f), which was then dried in an oven at 40 °C to obtain high-molecular-weight PDLLA in its dry form (Figure 3g).

Table 2 presents the initial weights of lactic acid used in the synthesis, along with the weights of the products obtained at the end of each stage, conducted in triplicate. The samples labeled PLA-1, PLA-2, PLA-3, and PLA-4—which correspond to an oligomerization time of 4 h—showed no evidence of polymer formation. In contrast, PLA-5, PLA-6, PLA-7, and PLA-8, synthesized with an oligomerization time of 6 h, yielded solid products, indicating successful progression of the reaction. The shorter oligomerization time likely limited the formation of linear oligomers with sufficient chain length and reactivity for the subsequent depolymerization and ring-opening reactions. As a result, the monomers generated were insufficient or not adequately structured for efficient ring-opening polymerization, leading to the formation of PDLLA with significantly lower molecular weight. This suggests that the oligomerization time variable plays a critical role in achieving suitable intermediates for the synthesis of high-molecular-weight PDLLA.

### 3.3. Characterization of PDLLA

The FTIR spectrum of the standard PDLLA is shown in Figure 4, displaying the characteristic absorption peaks corresponding to the functional groups that define the internal structure of PLA, thereby confirming the identity of the reference material. Figure 5 presents the FTIR spectra of the synthesized samples. The spectra were processed using OPUS software, developed for data analysis with the Bruker ALPHA II equipment, allowing a direct comparison between the synthesized PDLLA and the standard reference (CAS 26969-66-4).

For capillary viscometry, four polymer solutions were prepared for each sample at different concentrations: 0.3%, 0.5%, 0.7%, and 1%, using high-purity chloroform as the solvent. The flow times of the pure solvent and the polymer solutions were then measured using a stopwatch. Equation (1) was applied to calculate the reduced viscosity (η_red_), followed by a graphical determination of the intrinsic viscosity. Using Equation (2), the molecular weight of each of the 12 samples was determined. Table 3 presents the molecular weights of the 12 samples obtained, as well as the coefficient of variation for each sample code, all of which are below 5%. The molecular weight order in descending sequence was PLA-7 < PLA-8 < PLA-5 < PLA-6. The highest molecular weight was observed for sample PLA-7, exceeding 100 kDa, classifying it as high molecular weight. This result is consistent with the previous FTIR analysis.

To confirm the chemical structure of the synthesized polymer and assess the potential presence of impurities arising from the dimer synthesis, ^1^H-NMR analysis was performed. Figure 6 shows the spectrum of sample PLA-7, recorded at 500 MHz in CDCl_3_ at 25 °C. The characteristic signals of PLA were observed: the methine protons (CH, signal a) of the main chain appear at 5.18 ppm, while the methyl protons (CH_3_, signal b) are detected at 1.58 ppm. The integral ratio between these signals is 1:3, consistent with the expected stoichiometry of the repeating polymer units. Additionally, minor signals at 4.38 ppm and 1.47 ppm correspond to the terminal CH (signal c) and CH_3_ (signal d) groups, respectively. The 1:3 ratio is also preserved in these signals, supporting their structural assignment. The absence of additional resonances in aromatic or unexpected aliphatic regions, together with the low relative intensity of the terminal signals, indicates that no significant impurities—such as residual oligomers, free lactic acid, or water—were detected. These findings are in agreement with the high molecular weight determined by viscometry, which exceeded 100 kDa for PLA-7, supporting an efficient polymerization process.

In each of the 12 DSC diagrams, the glass transition temperature (Tg) of PLA should be between 49 and 56 °C, represented by the first peak (Zuluaga, 2013) [21]. The melting temperature (Tm) and the enthalpy of fusion (ΔHm) are represented by the second peak, which for the polymer studied is found between 132 and 146 °C (Boua-In et al., 2010) [12]. On the other hand, the crystallinity percentage (χ) is calculated from the ΔHm values using Equation (3). These results, along with Tg and Tm, are shown in Table 4. As observed, the thermal properties are directly proportional to the molecular weight, as stated by Zuluaga (2013) [21] and Pholharn et al. (2017) [18].

Through TGA, the thermal stability of PDLLA within a specific temperature range was determined, along with its degradation temperature. Since the resulting polymer is used in the preparation of a biomedical scaffold, it is essential that it does not degrade significantly at temperatures close to body temperature, approximately 37 °C. It is acknowledged that the degradation of PLA-based scaffolds in the human body is not governed solely by thermal effects. Physiological conditions, such as pH, ionic strength, and the presence of biological fluids, contribute to hydrolytic degradation and polymer erosion. However, this section is specifically focused on the thermal behavior of the material as characterized by TGA, which, while not representative of in vivo degradation, provides valuable information about its thermal stability and processing window. Table 5 presents the degradation temperatures (Td) of each sample. In general, Figure 7 shows that PLA-7 has the lowest degradation temperature but remains the most thermally stable at the desired temperature range.

### 3.4. Statistical Analysis

Although a full factorial design of 2^3^ was initially proposed, the results obtained from the samples corresponding to 4 h of oligomerization time (PLA-1 to PLA-4) showed that no polymer of sufficient molecular weight was formed. Consequently, these conditions were excluded from the subsequent statistical analysis. Therefore, the statistical treatment presented below corresponds to a reduced factorial design of 2^2^, based on the successful synthesis results obtained with 6 h of oligomerization.

To apply analysis of variance (ANOVA), the data must meet three key assumptions: (i) normal distribution, (ii) independence, and (iii) homogeneity of variances. Therefore, the corresponding tests were conducted. The Anderson–Darling test statistic yielded a value of 1.265. Since the *p*-value (0.016) is lower than the significance level (α = 0.05), it is likely that the data do not follow a normal distribution; however, this conclusion could be influenced by the small sample size.

According to the Durbin–Watson test for independence, it can be stated that the data meet the independence assumption, as the obtained test statistic value (d = 1.9266) falls within the acceptable range of 1.5–2.5. Levene’s test statistic was used to assess homoscedasticity, i.e., whether variances are equal across groups. Given that χ^2^ is the critical value of the chi-square distribution with 3 degrees of freedom and a significance level of 0.05, and since T (3.2438) is lower than χ^2^ (7.8147), it is established that the variances are equal.

Among the three ANOVA assumptions analyzed (Figure 8), independence of residuals and equality of variances were satisfied. Despite the normality test indicating that the data do not follow a normal distribution, ANOVA was still applied, considering that this method is robust to certain violations of normality, especially when working with small samples.

Table 6 presents the results of the ANOVA. By analyzing the *p*-values of the sources of variation, it was determined that the two variables studied—ROP time and catalyst concentration—as well as their interaction (ROP time × catalyst concentration), have a statistically significant effect on the response variable, namely, the molecular weight of the synthesized PDLLA, since *p* < 0.05. This result is supported by the trends observed in Figure 9 and Figure 10.

Figure 10 presents the Pareto chart of standardized effects for the molecular weight. This chart helps identify which variables have a statistically significant impact on the response, based on the magnitude of their standardized effects. The dotted reference line at 2.3 represents the threshold for statistical significance at a significance level of α = 0.05. Any effect that extends beyond this line can be considered statistically significant, indicating that the corresponding variable or interaction has a measurable influence on the molecular weight of the synthesized PDLLA.

In this case, the bars corresponding to ROP time, catalyst concentration, and their interaction exceed the 2.3 threshold, confirming their significance in the model. These results are consistent with the *p*-values obtained in the ANOVA (Table 6), reinforcing that the mentioned factors play a crucial role in determining the final molecular weight. The Pareto chart thus provides a clear visual complement to the statistical analysis, highlighting the relative importance of each variable.

Equation (4) expresses the mathematical relationship between ROP time and catalyst concentration to predict the molecular weight of PDLLA, with the oligomerization time fixed at 6 h.(4)$$\mathrm{Masa}\;\mathrm{molecular}\;(\bar{M})=-265,650+76,575\times X_{1}+110,173\times X_{2}-30,816\times X_{1\times 2}$$
where X_1_ represents the ROP time and X_2_ represents the catalyst concentration.

The software evaluates data reliability via a model fitting process that captures dataset dispersion and compares observed versus predicted values. The regression model is statistically robust, accounting for nearly all variability in the data. In particular, the R^2^ value of 99.95% indicates an exceptionally strong correlation, with experimental points clustering tightly around the regression line. The adjusted R^2^ of 99.93% corrects for the number of predictors, mitigating overfitting concerns. Although slightly lower, the predicted R^2^ of 99.89% remains very high, demonstrating the model’s ability to generalize to new observations. Collectively, these metrics confirm the model’s reliability for interpreting the current dataset and forecasting future outcomes.

Figure 11 presents the contour plot that illustrates the relationship between the molecular weight of the synthesized PDLLA and the variables ROP time and catalyst concentration. In this analysis, the oligomerization time was fixed at 6 h, since polymer synthesis was not achieved at 4 h. The color contours in the graph represent different molecular weight ranges, from values below 20,000 Da (in dark blue) to values above 120,000 Da (in dark green). It is observed that increasing the ROP time and decreasing the catalyst concentration led to higher molecular weight values. This visualization provides a useful guide to understanding the influence of these variables within the studied range.

### 3.5. Kinetic Study

Table 7 presents the evolution of the polymer’s molecular weight over time at different reaction temperatures. It is observed that the initial molecular weight is zero and progressively increases as the reaction progresses. However, the rate of increase varies significantly with temperature, exhibiting behavior dependent on the reaction kinetics. At 180 °C, the molecular weight reaches its maximum value in a shorter time compared to the other temperatures, suggesting a higher polymerization rate at elevated temperatures. In contrast, at 150 °C, the increase in molecular weight is more gradual, indicating slower kinetics. This behavior is corroborated in Figure 12, where the curve corresponding to 180 °C shows a steeper slope during the initial minutes of the reaction, subsequently stabilizing at approximately 145 kDa. The overall trend suggests that the polymerization process follows a temperature-dependent growth model, with an initial phase of rapid molecular weight increase followed by a stabilization phase. This behavior can be attributed to monomer availability and the influence of temperature on the reaction rate and the mobility of oligomers within the system.

To determine the reaction order of poly-L-lactic acid (PLA) polymerization, the experimental data were analyzed using the integrated rate law for different reaction orders. As shown in Equation (5), the general equation for a first-order reaction is provided by the following expression.(5)$$\mathrm{Ln}\left[M\right]=-kp\times t+{Ln\left[M\right]}_{0},$$
where [M]_0_ is the initial monomer concentration, [M] is the monomer concentration at time t, and kp is the reaction rate constant.

The residual monomer concentration [M] is inversely proportional to the molecular weight (M_t_), allowing the equation to be reformulated in terms of M_t_. As shown in Equation (6), the relationship is expressed in the following mathematical form:(6)$$\mathrm{Ln}\left[\frac{M_{t}}{M-M_{t}}\right]=kp\times t+C,$$
where M represents the maximum molecular weight attainable under the given system conditions.

Figure 13 presents the plot of Ln(Mt/(M − Mt)) as a function of time. A linear fit with high correlation is observed, indicating that the reaction follows first-order kinetics. This suggests that the reaction rate depends directly on the monomer concentration. The slope of this line allows the determination of the rate constant kp, whose values are presented in Table 8 for the different conditions evaluated.

The obtained fit supports the hypothesis that the rate-determining step of the process is the incorporation of new monomeric units into the growing chain. Moreover, the linearity of the fit and the consistency of kp values under different experimental conditions reinforce the validity of the employed kinetic model. The rate constant exhibits a systematic variation with temperature, suggesting that temperature plays a crucial role in the polymerization kinetics. The analysis of kp values as a function of temperature reveals an increase in reaction rate with rising temperature, in agreement with the Arrhenius theory. This behavior is characteristic of systems where activation energy controls the process rate.

These results enable the prediction of the temporal evolution of the reaction and the evaluation of the impact of experimental conditions on the system’s kinetics, providing key insights into the PLA polymerization mechanism. The validation of the first-order model and the determination of kp confirm the applicability of this approach for describing the process and contribute to a better understanding of how different variables influence polymer synthesis under the tested conditions.

Figure 14 presents the graphical representation of Ln(kp) as a function of 1/T, according to Equation (7):(7)$$\mathrm{Ln}\left(kp\right)=-\frac{Ea}{R}\cdot \frac{1}{T}+\mathrm{Ln}\left(A\right)$$
where kp is the rate constant, A is the pre-exponential factor, Ea is the activation energy, R is the ideal gas constant, which is 8.314 J/(mol·K), and T is the absolute temperature.

The linear fit obtained with a coefficient of determination (R^2^ = 0.9918) confirms the validity of the Arrhenius model in describing the polymerization kinetics of PLA. From the slope of the fitted line, the activation energy was determined to be Ea = 70.5 kJ/mol, indicating that the polymerization process requires overcoming a significant energy barrier for monomer incorporation into the polymer chain. The pre-exponential factor A, calculated from the intercept, was found to be 6.93 × 10^6^ min^−1^, suggesting a relatively high frequency of effective collisions between reactive species in the system.

The obtained Ea value is characteristic of ring-opening polymerization reactions, where temperature influences the stability of reactive intermediates and the mobility of species in the system. The linearity of the Arrhenius plot confirms that the reaction rate dependence on temperature follows the expected exponential trend for thermally activated chemical processes. The obtained Ea value is characteristic of ring-opening polymerization reactions, where temperature influences the stability of reactive intermediates and the mobility of species in the system. The linearity of the Arrhenius plot confirms that the reaction rate dependence on temperature follows the expected exponential trend for thermally activated chemical processes. These results provide insights into the reaction kinetics under varying thermal conditions and establish a basis for adjusting synthesis parameters to achieve better control over the molecular weight distribution of the resulting polymer.

### 3.6. Fabrication of a Polymeric Scaffold

The PLA-7 polymer, which exhibited the highest molecular weight and the lowest degradation around 37 °C, was selected for the preparation of the biomedical scaffold, for which 100 g were produced. During extrusion, as the 1.02 mm diameter filament was obtained, it was wound onto a spool to facilitate subsequent handling. The scaffold was designed using OpenSCAD version 2021.01, as shown in Figure 15a, and then segmented with Ultimaker Cura version 4.13.1 (Figure 15b). The obtained filament was then inserted into the Kingroon3D printer (Kingroon3D Technology Co., Ltd., Shenzhen, China).

Multiple trials were conducted to identify the best printing parameters, adjusting temperature and layer height. After seven tests, the results of which are shown in Figure 16, the ideal values were determined; these are detailed in Table 9.

The scaffold printed with a layer height of 0.04 mm and a temperature of 218 °C was selected due to its firmness in the traces, as shown in Figure 17. The scaffold was obtained in triplicate (S01, S02, and S03). Additionally, it is important to highlight that an adhesive was applied to the print bed to improve the adhesion of the printed piece.

### 3.7. Characterization of Scaffolds

The architecture, surface morphology, and structural stability of the scaffolds printed under the best-performing conditions are shown in Figure 18. Notably, although the samples were printed using the same parameters, the images reveal variations in surface structure and pore distribution. In Figure 18a (S01), a more homogeneous structure with a relatively uniform pore distribution is observed, suggesting consistency in the printing process and fiber formation. In contrast, Figure 18b (S02) exhibits slight variations in surface morphology, with areas where the pores appear more elongated or deformed, which could indicate local variations during the printing process or differences in fiber tension during solidification. Figure 18c (S03) presents an intermediate structure, displaying features observed in both S01 and S02, reinforcing the notion that, despite maintaining constant printing parameters, intrinsic variations exist in the microstructure of the scaffold. These variations may be attributed to factors such as material distribution, referring to how the material is deposited and organized during the scaffold printing process, including how it is extruded and layered, potentially influencing the uniformity of the final structure. Additionally, cooling dynamics, which describe how the printed material cools and solidifies after deposition, may lead to differences in material contraction or fiber formation during solidification. The rate and uniformity of cooling can affect pore formation, fiber stability, and the integration of the scaffold’s layers.

An average porosity of 71.6% was obtained, which falls within the porosity range of bone, varying between 50% and 90%. Highly porous structures are more suitable for facilitating the flow of nutrients and oxygen, enabling cell survival throughout the material [6]. The specific porosity values obtained for each scaffold are presented in Table 10.

Figure 19 illustrates the curve generated between the applied tensile force (MPa) and strain (%) of the printed scaffolds. Although all scaffolds were printed using the same parameters, slight variability in their mechanical behavior is observed. Scaffold S03 exhibits a higher strain profile compared to S01 and S02 for the same applied force, which could indicate lower stiffness or a greater capacity for deformation before reaching the same force as the other two scaffolds. On the other hand, S01 and S02 display very similar curves, suggesting consistency in their mechanical behavior, albeit with slight differences that could be attributed to intrinsic variations in the printing process or minor differences in the material’s microstructure.

These results highlight the importance of considering the inherent variability in scaffold manufacturing, even when using identical printing parameters. Table 11 provides mechanical compression test data for the three scaffold samples (S01, S02, S03). As shown, all scaffolds have similar area and thickness. The maximum compressive strength of trabecular bone ranges from 2 to 12 MPa, while its elastic modulus varies between 50 and 500 MPa [6]. Comparing these values with the results obtained from the three printed scaffolds—manufactured under identical parameters—it is observed that only the elastic modulus of S02 falls within the range of trabecular bone. However, the maximum compressive strength of all three scaffolds closely approximates that of trabecular bone, though with variations that could be attributed to subtle differences in the microstructure resulting from the printing process. This suggests that, while the scaffolds exhibit adequate compressive strength, further adjustment of the printing process parameters may be necessary to ensure greater consistency in the elastic modulus and enhance similarity to the mechanical properties of trabecular bone.

Table 12 presents the measured contact angles, along with the mean and standard deviation for each scaffold (S01, S02, S03). The average contact angle is 85.97°, with a relatively low standard deviation of 3.56° for all scaffolds. This low standard deviation indicates consistency in the measurements acquired at different positions of each sample. Since the average contact angle for the scaffolds is less than 90°, it can be concluded that the scaffolds exhibit hydrophilic behavior. This suggests good cell adhesion and enhanced liquid diffusion, characteristics that are favorable for tissue engineering applications.

## 4. Conclusions

The synthesis of poly-DL-lactic acid (PDLLA) via ring-opening polymerization (ROP) was successfully carried out, resulting in high-molecular-weight PDLLA. The experimental results indicate that an oligomerization time of 6 h, combined with a polymerization time of 4 h and a catalyst concentration of 1%, led to favorable conditions for the formation of long polymer chains. In contrast, a shorter oligomerization time of 4 h was insufficient to support adequate chain development, highlighting the importance of preliminary oligomer length in the overall polymerization process.

The characterization of the polymer was conducted using four key techniques that collectively enabled a comprehensive evaluation of its properties. FTIR confirmed the chemical structure of PDLLA by identifying the characteristic peaks of PLA. In parallel, proton nuclear magnetic resonance (^1^H-NMR) analysis of the PLA-7 sample confirmed the expected chemical structure of the polymer, revealing only the characteristic signals of PLA without evidence of residual monomer, oligomers, or other impurities. Likewise, capillary viscometry revealed that the PLA-7 sample exhibited the highest molecular weight, exceeding 100 kDa. Additionally, DSC analysis showed that PLA-7 displayed the best thermal properties, with a Tg of approximately 52.77 °C and a Tm close to 143.57 °C. Finally, thermogravimetric analysis confirmed the thermal stability of PLA-7 at temperatures below 40 °C. All these characterizations collectively demonstrate that the synthesized polymer is suitable for applications in tissue engineering.

The kinetic study confirmed that PLA polymerization follows first-order reaction kinetics, with the reaction rate directly dependent on monomer concentration. The rate constant (kp) increased systematically with temperature, ranging from 0.0138 min^−1^ at 150 °C to 0.0500 min^−1^ at 180 °C, highlighting the significant influence of thermal conditions on the polymerization process. Furthermore, the Arrhenius analysis yielded an activation energy (Ea) of 70.5 kJ/mol, indicating that monomer incorporation into the growing polymer chain requires overcoming a considerable energy barrier. These findings provide valuable insights into the reaction mechanism, allowing for improved understanding and control of the synthesis parameters, with the aim of tailoring the molecular weight distribution and enhancing the potential applicability of PLA in biomedical and industrial contexts.

The statistical analysis conducted in this study was essential for identifying and understanding the influence of synthesis parameters on the properties of PDLLA. Using a factorial experimental design, the effects of three key variables were evaluated: oligomerization time, ROP reaction time, and catalyst concentration. The ANOVA results showed that these variables, as well as their interactions, have a significant impact on the polymer’s molecular weight. Moreover, a mathematical model was used to predict molecular weight as a function of these parameters, achieving a high reliability level with a coefficient of determination (R^2^) of 99.95%. This statistical approach provided valuable insights into the synthesis process and established a robust basis for future developments involving PDLLA in tissue engineering.

The scaffold, manufactured using 3D printing at a temperature of 218 °C with the FDM technique, was evaluated through four main characterizations. The porosity reached an average of 71.6%, which is suitable for facilitating nutrient permeability in tissue engineering applications. Regarding the contact angle, the scaffold surface exhibited an average value of 85.97°, indicating a hydrophilic behavior. Additionally, SEM images revealed variations in the microstructure. Finally, mechanical tests showed that the S02 scaffold had an elastic modulus of 51.7 MPa and a compressive strength of 4.24 MPa, falling within the range of human trabecular bone.

## Figures and Tables

**Figure 1 polymers-17-01708-f001:**
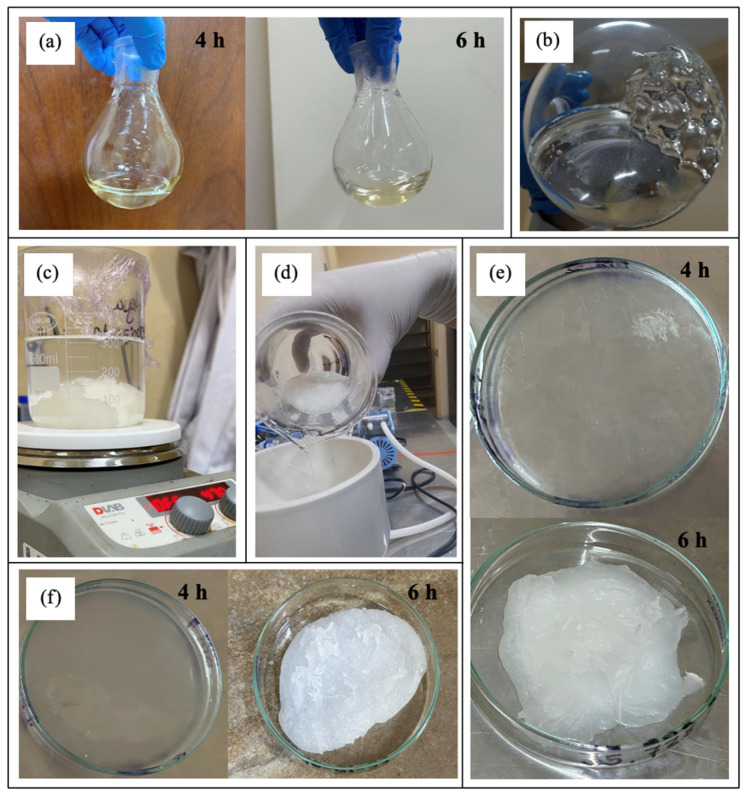
Process of PDLLA oligomerization. (**a**) Product at the end of the reaction. (**b**) Dissolution of the formed product. (**c**) Precipitate formation. (**d**) Vacuum filtration for precipitate recovery. (**e**) Oligomer placed in a Petri dish for subsequent drying. (**f**) Dried oligomer.

**Figure 2 polymers-17-01708-f002:**
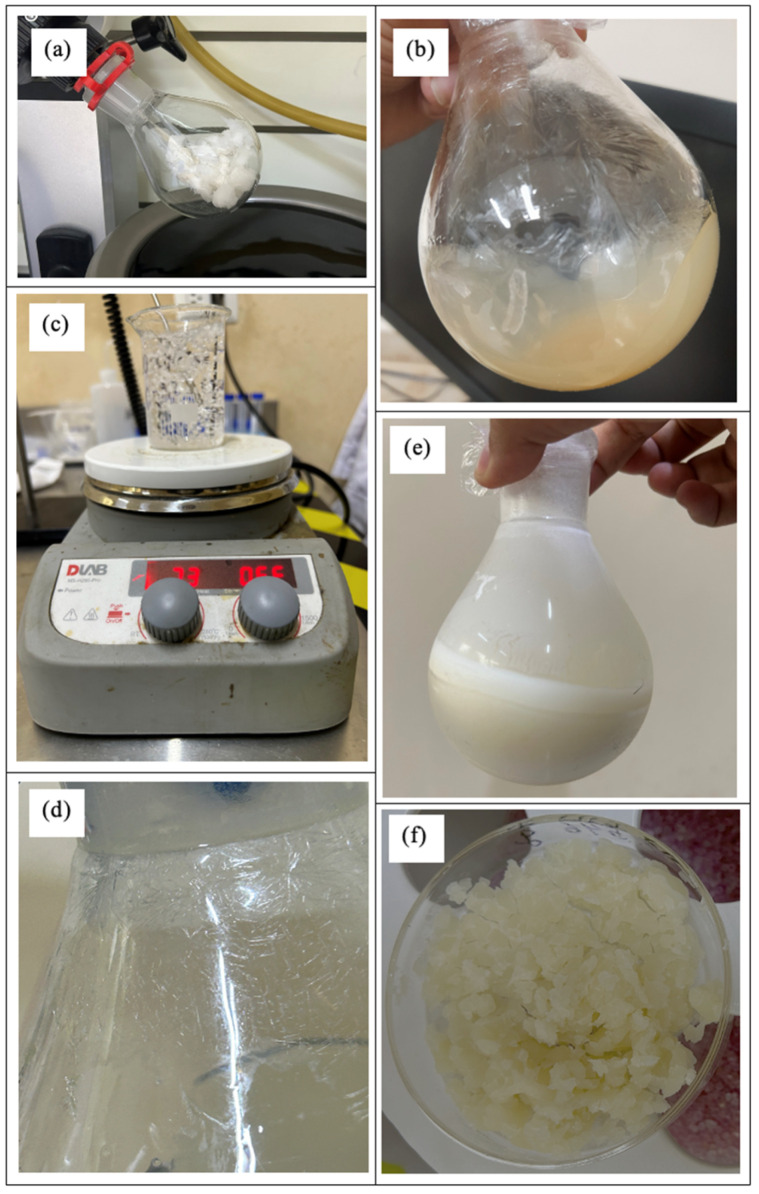
Process of PDLLA depolymerization. (**a**) Oligomer used, obtained in the previous stage. (**b**) Reaction product. (**c**) Boiling ethyl acetate. (**d**) Onset of lactide crystallization. (**e**) Lactide after refrigeration. (**f**) Dried lactide.

**Figure 3 polymers-17-01708-f003:**
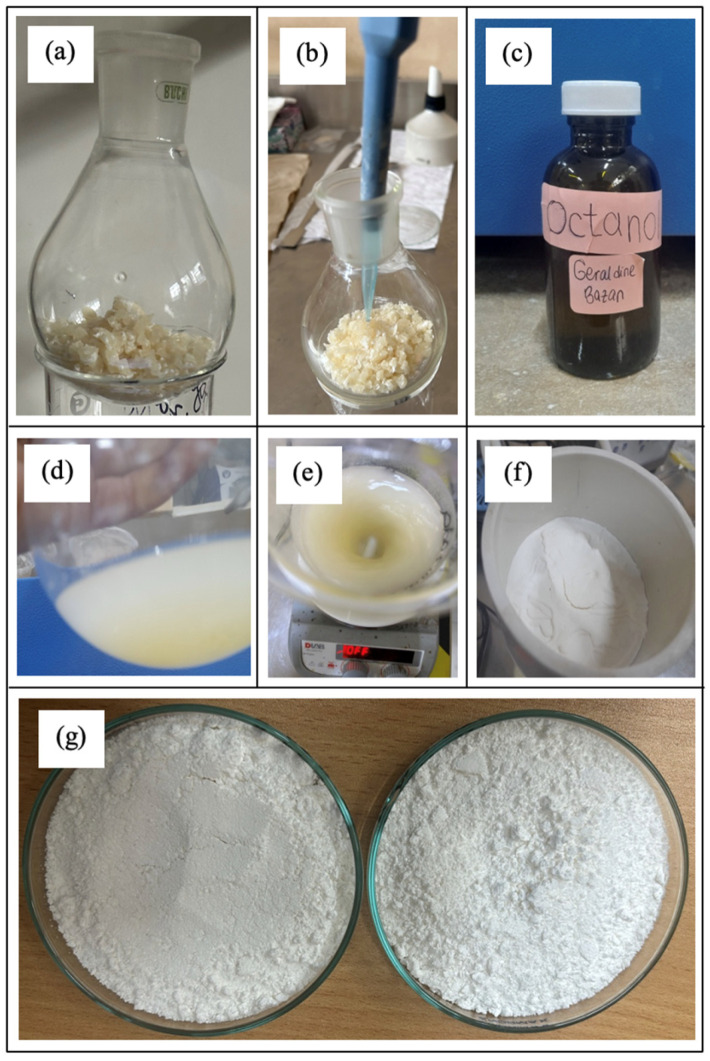
Process of ring-opening polymerization. (**a**) Dried lactide used for the reaction. (**b**) Catalyst addition. (**c**) Initiator. (**d**) Dissolution in chloroform. (**e**) Precipitation in methanol. (**f**) Vacuum filtration result. (**g**) Dried high-molecular-weight PDLLA.

**Figure 4 polymers-17-01708-f004:**
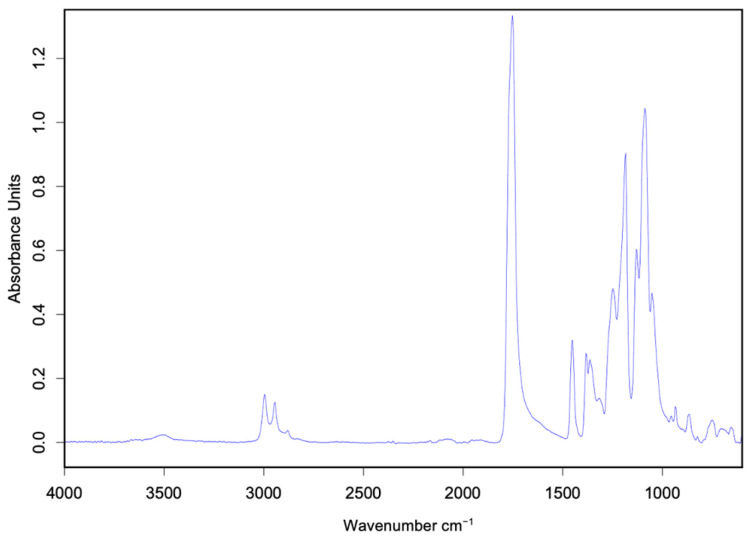
FTIR spectrum of standard PDLLA.

**Figure 5 polymers-17-01708-f005:**
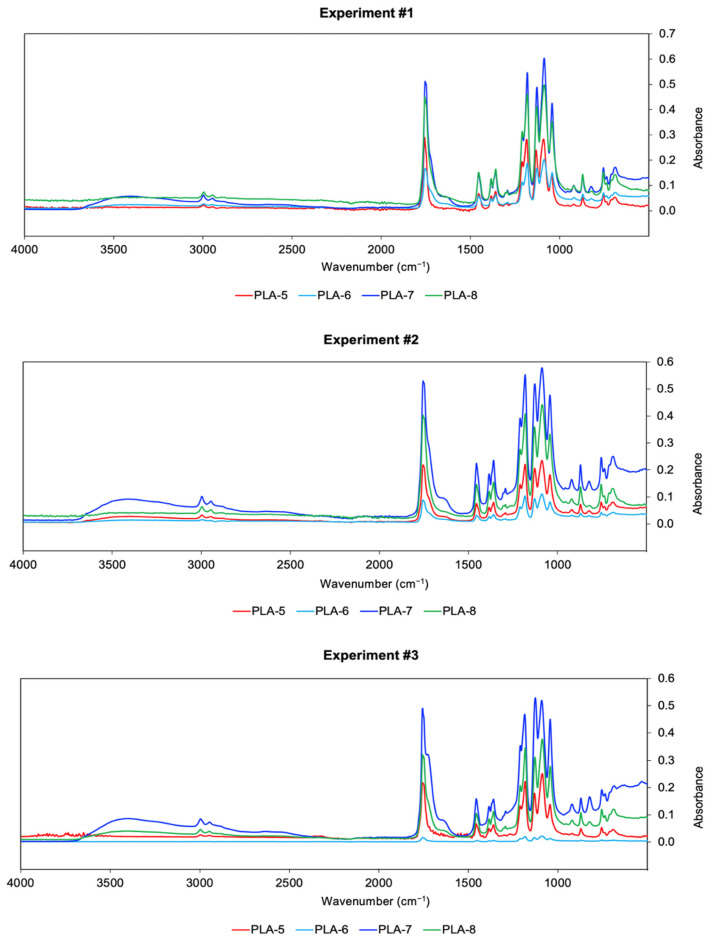
FTIR spectrum of PDLLA samples.

**Figure 6 polymers-17-01708-f006:**
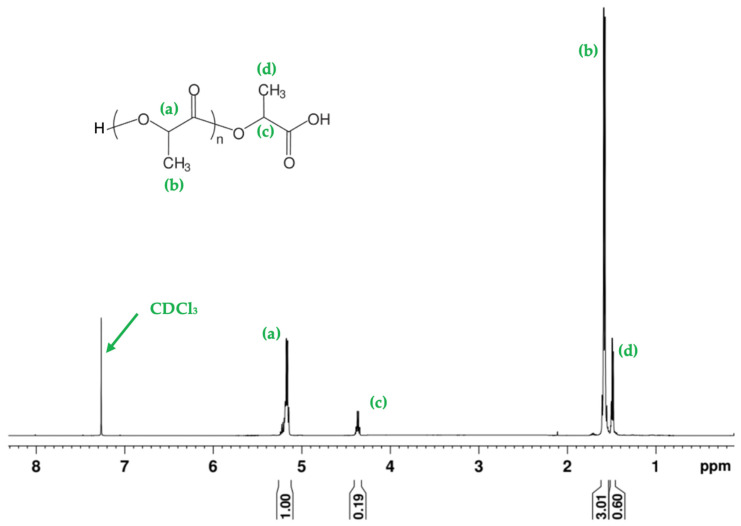
The ^1^H-NMR spectrum of PLA-7 sample recorded at 500 MHz in CDCl_3_ at 25 °C. The characteristic signals observed include: (a) methine protons (CH) of the polymer backbone at 5.18 ppm, (b) methyl protons (CH_3_) of the repeating units at 1.58 ppm, (c) methine proton (CH) of the terminal group at 4.38 ppm, and (d) methyl protons (CH_3_) of the terminal group at 1.47 ppm.

**Figure 7 polymers-17-01708-f007:**
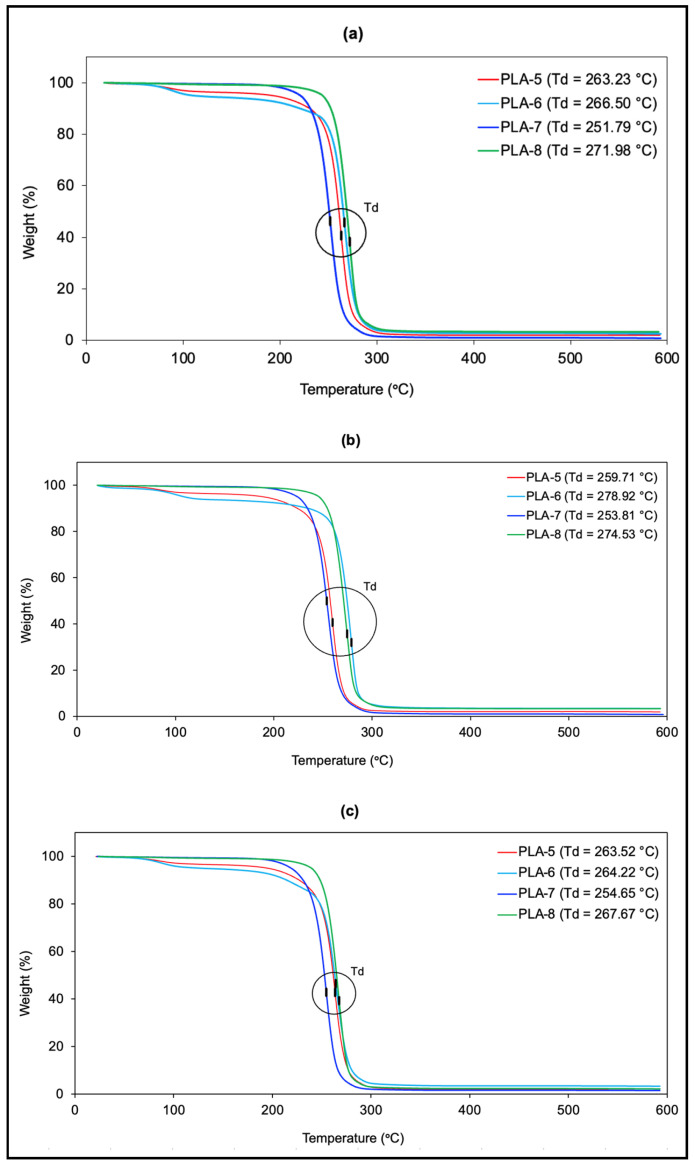
TGA diagrams. (**a**) Experiment #1. (**b**) Experiment #2. (**c**) Experiment #3.

**Figure 8 polymers-17-01708-f008:**
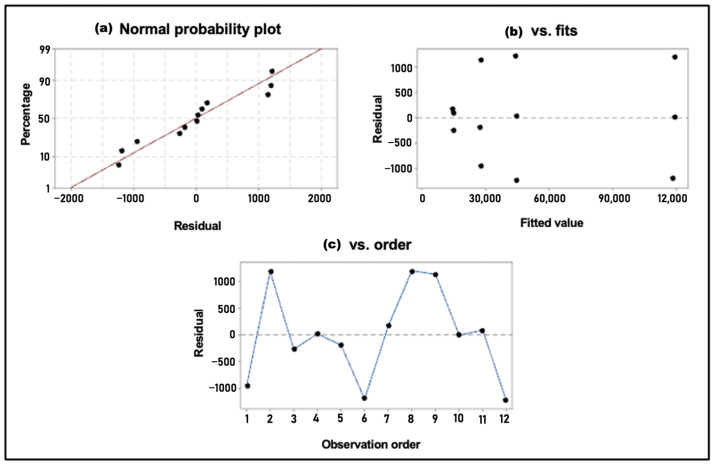
Residual plot for molecular weight. The figure displays three diagnostic plots used to evaluate the assumptions of ANOVA: (**a**) normal probability plot, where each dot represents a standardized residual; (**b**) residuals versus fitted values, showing the spread of residuals relative to predicted values; and (**c**) residuals versus observation order, where each point corresponds to a residual plotted in the sequence in which the data were collected. These plots are used to assess normality, homoscedasticity, and independence of residuals, respectively.

**Figure 9 polymers-17-01708-f009:**
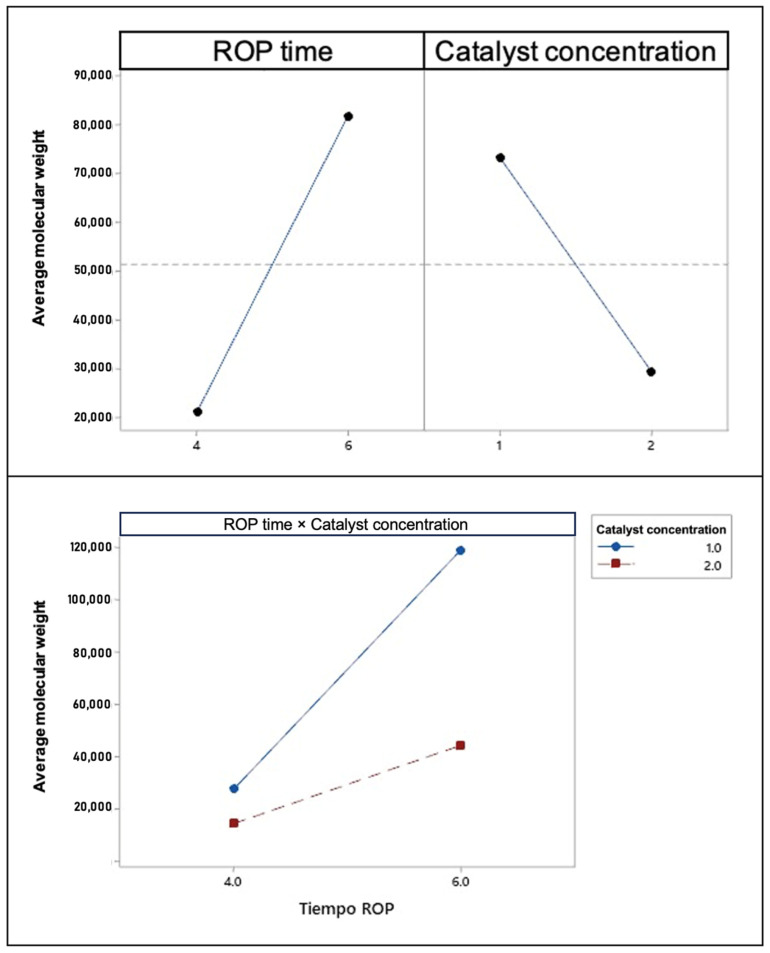
Main effects and interaction plot for molecular weight, where the dots represent the average molecular weight of PDLLA obtained under each experimental condition. The slope and direction of the lines indicate how each factor and their combination influence the molecular weight of the synthesized polymer.

**Figure 10 polymers-17-01708-f010:**
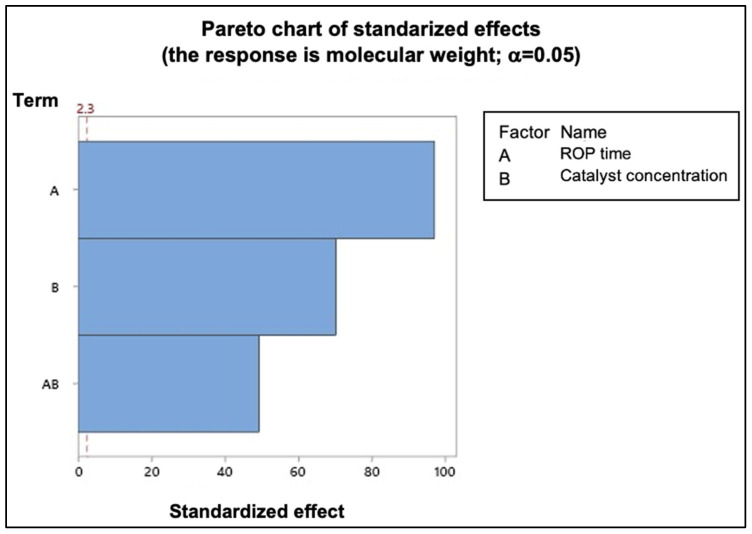
Pareto chart of standardized effects for molecular weight.

**Figure 11 polymers-17-01708-f011:**
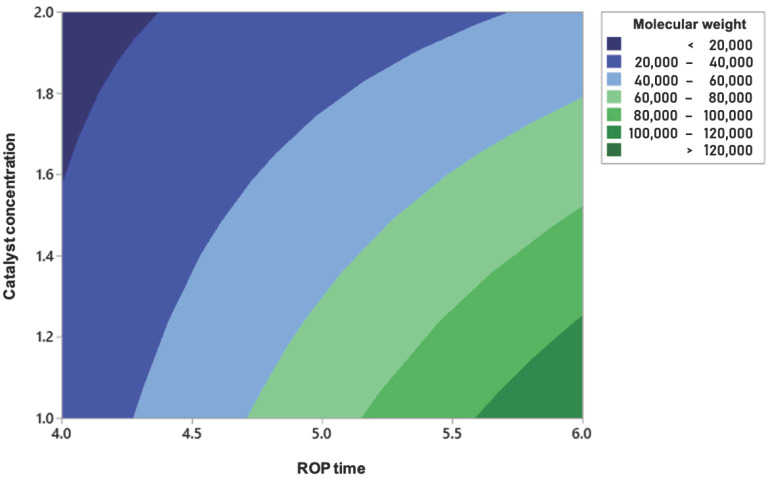
Contour plot.

**Figure 12 polymers-17-01708-f012:**
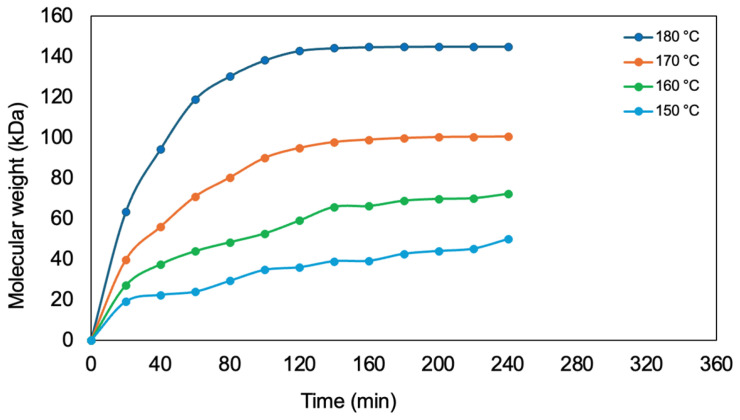
Molecular weight variation as a function of time at different reaction temperatures.

**Figure 13 polymers-17-01708-f013:**
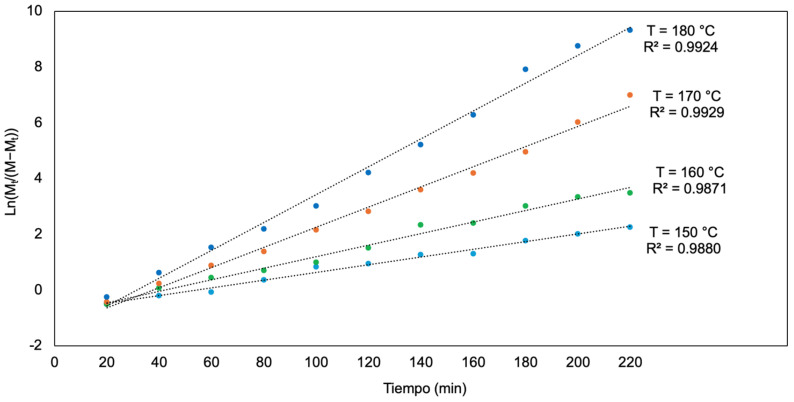
First-order kinetic plot for PLA polymerization: linear representation of Ln(M_t_/(M − M_t_)) as a function of time.

**Figure 14 polymers-17-01708-f014:**
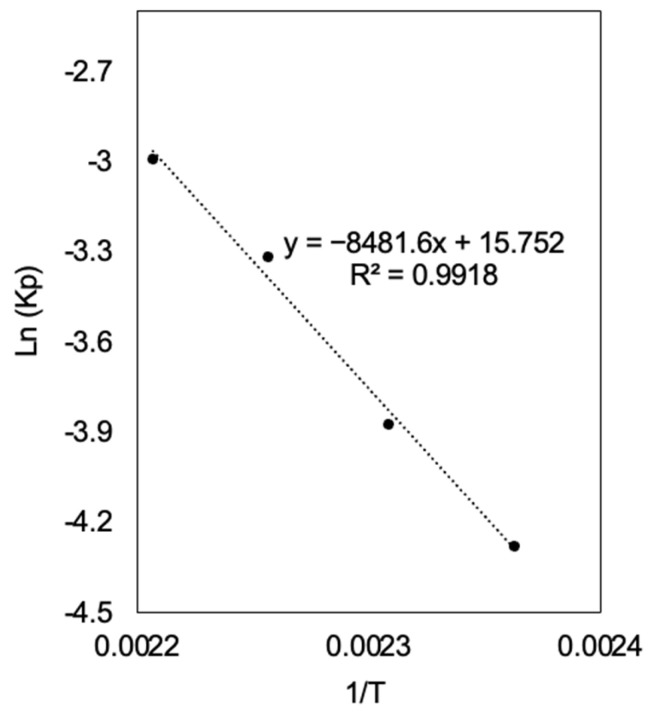
Arrhenius plot for the polymerization of PLA, showing the linear relationship between Ln(kp) and 1/T. The activation energy (Ea) and pre-exponential factor (A) were determined from the slope and intercept, respectively.

**Figure 15 polymers-17-01708-f015:**
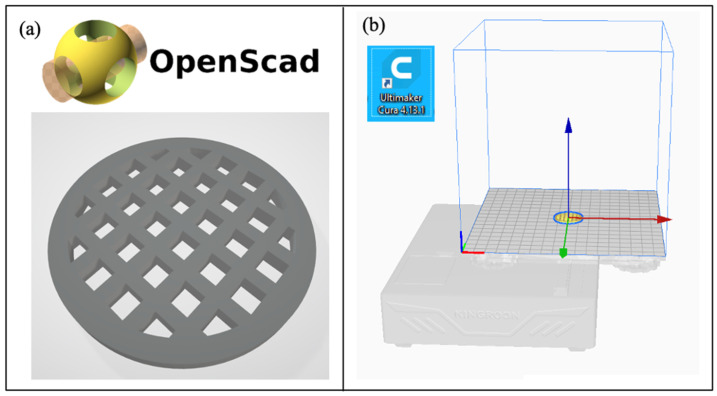
Scaffold. (**a**) Design using OpenSCAD software. (**b**) Segmentation using Cura software.

**Figure 16 polymers-17-01708-f016:**
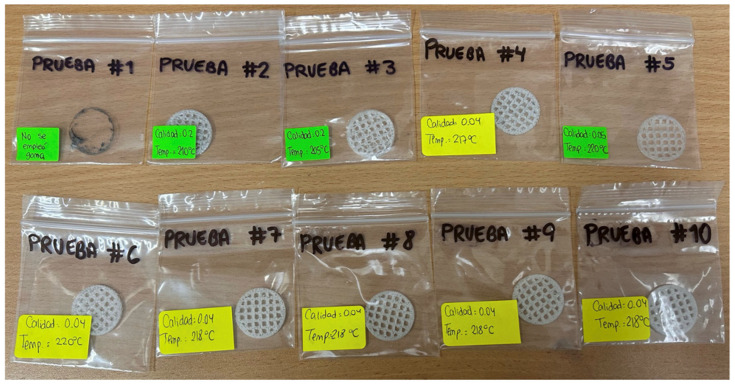
Three-dimensional printing tests.

**Figure 17 polymers-17-01708-f017:**
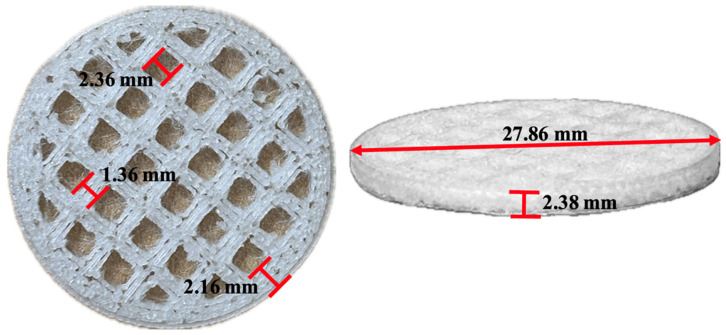
Three-dimension-printed PDLLA scaffold under the best-performing parameters.

**Figure 18 polymers-17-01708-f018:**
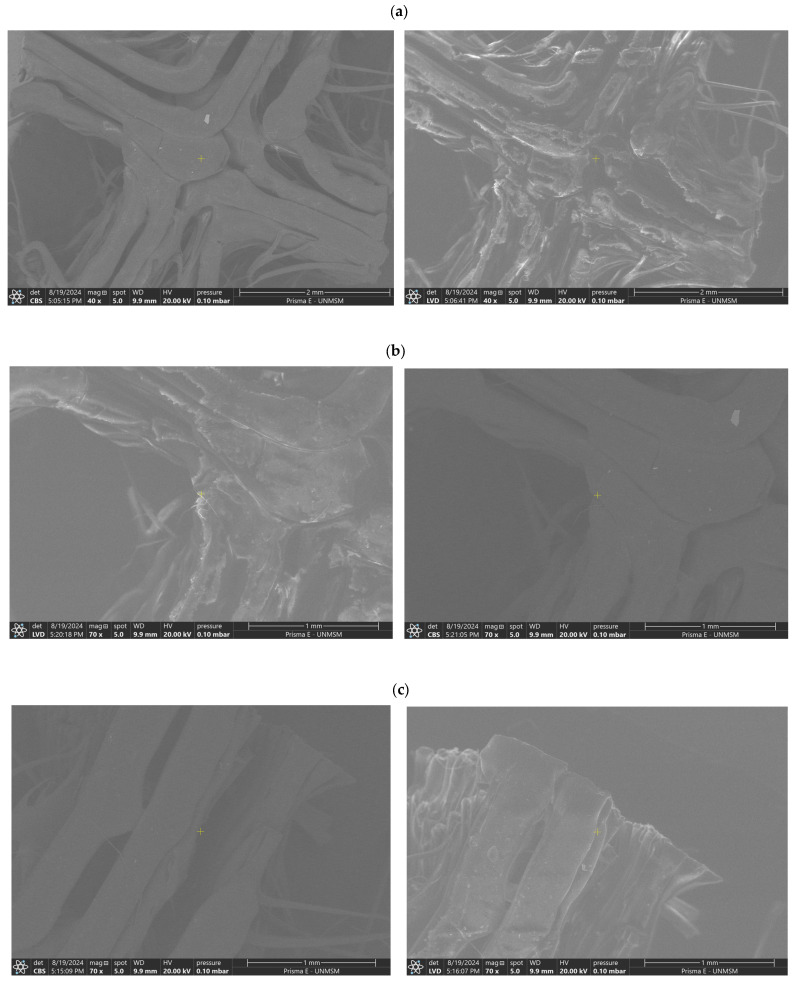
SEM images. (**a**) S01. (**b**) S02. (**c**) S03.

**Figure 19 polymers-17-01708-f019:**
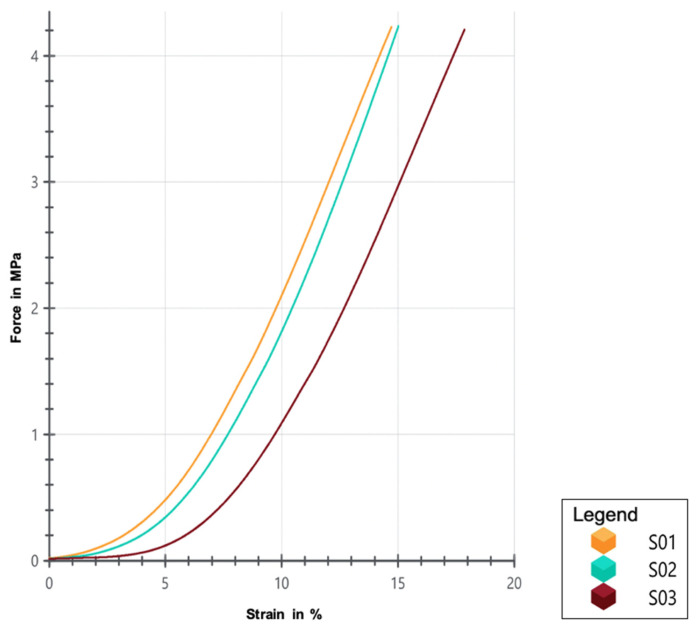
Force vs. strain curves for the scaffolds.

**Table 1 polymers-17-01708-t001:** Quantities of components used in the synthesis.

Sample Code	Oligomerization Time (h)	ROP Time (h)	% by Weight of Tin Octoate Catalyst
PLA-1	4	4	1
PLA-2	4	4	2
PLA-3	4	6	1
PLA-4	4	6	2
PLA-5	6	4	1
PLA-6	6	4	2
PLA-7	6	6	1
PLA-8	6	6	2

**Table 2 polymers-17-01708-t002:** Weights of lactic acid used at the beginning of the synthesis and of the products at the end of each stage.

Sample Code	Experiment Number	Lactic Acid Weight (g)	Dried Low-Molecular-Weight PDLLA Weight (g)	Dried Lactide Weight (g)	Dried High-Molecular-Weight PDLLA Weight (g)
PLA-1	1	60.543	11.298	2.272	-
	2	60.478	11.243	2.165	-
	3	60.450	11.145	2.122	-
PLA-2	1	60.456	11.221	2.140	-
	2	60.498	11.186	2.129	-
	3	60.470	11.200	2.137	-
PLA-3	1	60.557	11.320	2.251	-
	2	60.469	11.238	2.153	-
	3	60.478	11.239	2.159	-
PLA-4	1	60.502	11.189	2.128	-
	2	60.499	11.191	2.130	-
	3	60.489	11.187	2.128	-
PLA-5	1	60.449	38.440	15.716	4.964
	2	60.572	38.789	16.023	5.159
	3	60.461	38.549	16.234	5.767
PLA-6	1	60.448	38.096	15.844	4.427
	2	60.603	38.980	16.427	4.842
	3	60.588	39.097	16.234	4.631
PLA-7	1	60.462	38.773	16.126	10.503
	2	60.466	38.794	16.423	10.984
	3	60.453	38.726	16.121	10.558
PLA-8	1	60.557	39.350	16.387	7.837
	2	60.669	39.144	16.209	7.435
	3	60.477	38.143	15.987	7.234

**Table 3 polymers-17-01708-t003:** Average molecular weight per sample code.

Sample Code	Molecular Weight (g/mol)			Coefficient of Variation (%)	Average Molecular Weight (g/mol)
	Experiment #1	Experiment #2	Experiment #3		
PLA-5	26,781.24	26,944.24	28,958.42	4.4	27,561.30
PLA-6	14,386.82	14,211.91	14,814.55	2.1	14,471.10
PLA-7	120,457.89	117,452.10	119,331.80	1.3	119,080.60
PLA-8	44,563.81	45,142.18	43,370.76	2.0	44,358.92

**Table 4 polymers-17-01708-t004:** Thermal properties for each sample code concerning its molecular weight.

Sample Code	Molecular Weight (g/mol)	Tg (°C)	Tm (°C)	ΔHm (J/g)	χ (%)
PLA-5	27,561.30	50.16	136.89	43.73	46.67
PLA-6	14,471.10	49.16	133.45	41.80	44.61
PLA-7	119,080.60	52.77	143.57	44.87	47.88
PLA-8	44,358.92	52.26	143.28	44.37	47.35

**Table 5 polymers-17-01708-t005:** PDLLA degradation temperature.

Sample Code	Experiment Number	Degradation Temperature (°C)
PLA-5	1	263.23
	2	259.71
	3	263.52
PLA-6	1	266.50
	2	278.92
	3	264.22
PLA-7	1	251.79
	2	253.81
	3	254.65
PLA-8	1	271.98
	2	274.53
	3	267.67

**Table 6 polymers-17-01708-t006:** ANOVA for molecular weight.

Source	DF	Adjusted SS	Adjusted MS	F-Value	*p*-Value
Model	3	19,686,791,944	6,562,263,981	5598.35	0.000
ROP time	1	11,054,767,100	11,054,767,100	9430.95	0.000
Catalyst concentration	1	5,783,195,404	5,783,195,404	4933.71	0.000
ROP time × catalyst concentration	1	2,848,829,440	2,848,829,440	2430.37	0.000
Error	8	9,377,433	1,172,179		
Total	11	19,696,169,377			

**Table 7 polymers-17-01708-t007:** Evolution of molecular weight over time at different reaction temperatures.

	Temperature (°C)			
Time (min)	150 °C	160 °C	170 °C	180 °C
0	0	0	0	0
20	19.09	27.06	39.73	63.36
40	22.34	37.44	55.97	94.35
60	23.92	43.91	70.89	118.92
80	29.31	48.32	80.30	130.30
100	34.67	52.63	89.97	138.13
120	35.94	59.03	94.83	142.83
140	38.92	65.74	97.74	144.14
160	39.14	66.14	98.94	144.66
180	42.57	68.77	99.74	144.88
200	43.94	69.64	100.20	144.91
220	45.10	69.99	100.34	144.92
240	49.83	72.13	100.44	144.93

**Table 8 polymers-17-01708-t008:** Reaction rate constants (kp) determined for different temperatures.

Temperature (°C)	kp (min^−1^)
150	0.0138
160	0.0207
170	0.0362
180	0.0500

**Table 9 polymers-17-01708-t009:** Three-dimensional printing parameters.

Quality	
Layer height (mm)	0.04
Initial layer height (mm)	0.32
Line width (mm)	0.44
Top/bottom	
Top layers	8
Bottom layers	10
Top/bottom pattern	Lines
Infill	
Infill density (%)	60
Infill pattern	Trihexagonal
Connect infill lines	Yes
Material	
Printing temperature (°C)	218
Initial layer printing temperature (°C)	218
Build plate temperature (°C)	60
Flow (%)	100
Speed	
Print speed (mm/s)	60
Outer wall speed (mm/s)	30
Inner wall speed (mm/s)	30
Travel speed (mm/s)	150
Initial layer speed (mm/s)	20
Retraction	
Retraction distance (mm)	1
Retraction speed (mm/s)	40
Z-hop height (mm)	0.08
Cooling	
Fan speed (%)	100
Regular fan speed at height (mm)	0.4

**Table 10 polymers-17-01708-t010:** Data for the calculation of scaffold porosity.

	S-01	S-02	S-03
Dry scaffold mass (g)	0.506	0.516	0.498
Saturated scaffold mass (g)	0.800	0.820	0.790
Pore volume (cm^3^)	0.294	0.303	0.292
Apparent volume (cm^3^)	0.413	0.422	0.407
Porosity (%)	71.2	72.0	71.7
Average porosity (%)	71.6		

**Table 11 polymers-17-01708-t011:** Mechanical properties of the printed scaffolds.

	Area (mm^2^)	Thickness (mm)	Maximum Force (kN)	Maximum Stress (MPa)	Nominal Strain at Fmax (mm)	Nominal Strain at Fmax (%)	Elastic Modulus (MPa)
S01	591.8	2.20	2.50	4.23	0.324	14.7	45.7
S02	590.5	2.20	2.50	4.24	0.331	15.0	51.7
S03	594.4	2.20	2.50	4.21	0.393	17.9	43.4
Average				4.23	0.349	15.9	46.9

**Table 12 polymers-17-01708-t012:** Contact angles of the scaffolds.

Sample	Position 1	Position 2	Position 3	Average	Standard Deviation
S01	88.7°	86.4°	91.7°	88.93°	2.66°
S02	86.5°	79.2°	86.0°	83.90°	4.08°
S03	82.2°	86.5°	86.5°	85.07°	2.48°
Total average				85.97°	3.56°

## Data Availability

The data presented in this study are available within this article.

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
