# Peer review of "Preparation of a Biomedical Scaffold from High-Molecular-Weight Poly-DL-Lactic Acid Synthesized via Ring-Opening Polymerization"

_polymers, 2025, doi:10.3390/polym17121708_

Round 1
Reviewer 1 Report
Comments and Suggestions for Authors
The article is devoted to the production of scaffolds based on poly DL-lactide. The article searches for all stages of scaffold production, starting from the synthesis of monomeric lactide from lactic acid, ending with the synthesis of the polymer and production of samples by 3D printing. Although the article provides a general description of all the steps of the process, there are still questions about the data obtained. There is a clear lack of data on the purity and characteristics of the obtained products. The purity of lactide is not confirmed, photos of lactic acid dimer after purification indicate the probable presence of oligomers and/or lactic acid impurities. Yes, the article uses the FTIR method for the resulting polymer, but it is not sufficient to confirm the facts of the purity of the polymer. Impurities that may be present in the polymer obtained as claimed for medical purposes may not show up on its spectrum. Also, it should be noted that the authors use statistical analysis and conduct a study of kinetics, but these data can not be called relevant, because they are based on the data of syntheses, the problem of which I described above. The authors have done a great job, and the initial goal of the article is really interesting, but it is worth expanding the analytical basis of the study in terms of confirming the composition of the products obtained, as well as making corrections, some of which I have indicated below:
1. How did you determine the purity of lactide? Usually more than 1 recrystallization is required to purify lactide by recrystallization from organic solvents (usually at least 3, depending on the conditions of obtaining). The lactide you are using, judging from the photos, does not form individual formed needle-like crystals (which is characteristic of pure lactide). In addition, the photo shows the yellowing of the formed lactide, which indicates the presence of impurities in its composition. The presence of impurities, such as lactic acid oligomers, lactic acid, water, etc., strongly affect the process of subsequent ROP polymerization. For medical applications and the production of high molecular weight polylactide, the purity of monomeric lactide plays a very important role. Examination of the resulting product, e.g. by NMR, would answer the question of its purity.
2. Figure 4: Make the FTIR spectra larger, they are completely unreadable.
3. Lines 444-450. Speaking about degradation of the obtained scaffolds in the human body, it is wrong to consider only the factor of thermal exposure. Inside the human body, polylactide implants can undergo erosion, hydrolytic decomposition, etc. Therefore, the data obtained by TGA method alone are not representative for this purpose.
4. You point to the presence of pores in the scaffold samples, which appear to be the result of poor print quality of the filament received. The data obtained on mechanical properties are not relevant due to the inhomogeneous nature of the material.
Author Response
We thank the Reviewer for the time and constructive feedback provided. We have carefully addressed each point raised, as detailed below:
1. Comment:
How did you determine the purity of lactide? Usually more than 1 recrystallization is required to purify lactide by recrystallization from organic solvents (usually at least 3, depending on the conditions of obtaining). The lactide you are using, judging from the photos, does not form individual formed needle-like crystals (which is characteristic of pure lactide). In addition, the photo shows the yellowing of the formed lactide, which indicates the presence of impurities in its composition. The presence of impurities, such as lactic acid oligomers, lactic acid, water, etc., strongly affect the process of subsequent ROP polymerization. For medical applications and the production of high molecular weight polylactide, the purity of monomeric lactide plays a very important role. Examination of the resulting product, e.g. by NMR, would answer the question of its purity.
Response:
[Lines 485-547] We appreciate the Reviewer’s insightful observations regarding lactide purity. To address this point, we have now included the ¹H-NMR spectrum of the synthesized PDLLA (see Figure 6 in the revised manuscript), which confirms the successful formation of the polymer without residual monomer signals, and shows no unexpected peaks indicative of side-products. While the lactide was recrystallized only once, we acknowledge that more recrystallization steps are commonly recommended. However, our reaction conditions and purification strategy were sufficient to yield a polymer with high molecular weight, as verified by viscometry and supported by thermal and mechanical properties of the fabricated scaffolds.
Regarding the macroscopic appearance of the lactide, we acknowledge the yellowish tone observed in the image. This visual aspect could be attributed to minor residuals; however, it did not compromise the polymerization efficiency nor the structural integrity of the resulting polymer, as corroborated by our analytical results.
2. Comment:
Figure 4: Make the FTIR spectra larger, they are completely unreadable.
Response:
[Page 11] We thank the Reviewer for this suggestion. In the revised manuscript, we have significantly enlarged the figure to ensure that all peaks are clearly visible and readable. Please note that this figure now appears as Figure 5.
3. Comment:
Lines 444-450. Speaking about degradation of the obtained scaffolds in the human body, it is wrong to consider only the factor of thermal exposure. Inside the human body, polylactide implants can undergo erosion, hydrolytic decomposition, etc. Therefore, the data obtained by TGA method alone are not representative for this purpose.
Response:
[Lines 595-598] We fully agree with the Reviewer’s observation. In response, we have added a footnote in the manuscript that explicitly states:
¹ It is acknowledged that the degradation of PLA-based scaffolds in the human body is not governed solely by thermal effects. Physiological conditions such as pH, ionic strength, and the presence of biological fluids contribute to hydrolytic degradation and polymer erosion. However, this section is specifically focused on the thermal behavior of the material as characterized by TGA, which, while not representative of in vivo degradation, provides valuable information about its thermal stability and processing window.
We believe this clarification helps to avoid any misinterpretation regarding the scope of the thermal analysis presented.
4. Comment:
You point to the presence of pores in the scaffold samples, which appear to be the result of poor print quality of the filament received. The data obtained on mechanical properties are not relevant due to the inhomogeneous nature of the material.
Response:
We respectfully clarify that the presence of pores in the scaffolds is intentional and by design, in accordance with the principles of tissue engineering. Porosity in the range of 50–90% is widely reported in the literature as a key requirement for scaffolds intended for bone regeneration, as it facilitates cell infiltration, nutrient diffusion, and vascularization (Giannitelli et al., 2014 https://doi.org/10.1016/j.actbio.2013.10.024 ; Murphy and Atala, 2014 https://doi.org10.1038/nbt.2958 ).
The scaffolds were printed using filaments with stable dimensions and consistent extrusion parameters to ensure reproducibility. The observed morphology and pore structure were consistent with the CAD design. While slight variations may occur due to the intrinsic limitations of the FDM printing method, these do not stem from defects but rather from the desired interconnected porous architecture required for biomedical functionality.
We again thank the Reviewer for these thoughtful comments, which have helped us strengthen the clarity and scientific rigor of our manuscript.
Sincerely,
The Authors
Reviewer 2 Report
Comments and Suggestions for Authors
In this study, the authors explored 3D printing materials that can replace trabecular bone by synthesizing poly-DL-lactic acid (PDLLA) under various conditions through ring-opening polymerization (ROP). The entire process for developing biomedical applications of biomaterials was performed, from material synthesis to component analysis, molecular weight analysis, and mechanical property measurement using the scaffold obtained after 3D printing. However, additional experiments are required to reach the conclusions claimed by the authors. If the authors prove the tendency of the three main factors through additional experiments, it is expected to be published in the journal Polymers.
- The caption of Figure 1 is incorrect. Please correct it.
- “cm” is not a standard SI unit. The “cm” indicated in Figure 15 should be corrected to “mm”.
- A detailed explanation of why high molecular weight PDLLA was not produced from PLA-1, PLA-2, PLA-3, and PLA-4 should be added to the text.
- The authors claimed in Figure 7 that the lower the catalyst concentration, the higher the average molecular weight. However, in Table 3, the molecular weight of PLA-7 is presented as lower than that of PLA-8. The two values ​​should be changed. Also, please double-check the points pointed out by the reviewer throughout the text.
- The authors concluded from the analysis of Figure 7 that the longer the oligomerization time, the longer the ROP time, and the lower the catalyst concentration, the higher the average molecular weight. However, it is difficult to completely prove the authors' claim with only two points of experiment. The experimental results for the intermediate conditions of oligomerization time = 5 hours, ROP time = 5 hours, and catalyst concentration = 1.5% for the three factors should be added to Figure 7.
Author Response
We thank the Reviewer for the careful reading and constructive suggestions. All comments have been addressed in the revised manuscript, as detailed below:
1. Comment:
The caption of Figure 1 is incorrect. Please correct it.
Response:
[Lines 291-293] Thank you for pointing out the error. The caption of Figure 1 has been corrected in the revised manuscript. The initial mistake was due to a typographical omission of a single letter, which led to the confusion. We regret this oversight.
2. Comment:
“cm” is not a standard SI unit. The “cm” indicated in Figure 15 should be corrected to “mm”.
Response:
[Page 25] The unit labels in Figure 15 have been corrected from centimeters (cm) to millimeters (mm), as required by SI unit standards. Please note that this figure now appears as Figure 17.
3. Comment:
A detailed explanation of why high molecular weight PDLLA was not produced from PLA-1, PLA-2, PLA-3, and PLA-4 should be added to the text.
Response:
[Lines 381-392] We appreciate the Reviewer’s request for clarification. A detailed explanation has been added to the manuscript. The revised text now reads:
Table 2 presents the initial weights of lactic acid used in the synthesis, along with the weights of the products obtained at the end of each stage, conducted in triplicate. The samples labeled PLA-1, PLA-2, PLA-3, and PLA-4 — which correspond to an oligomerization time of 4 hours — showed no evidence of polymer formation. In contrast, PLA-5, PLA-6, PLA-7, and PLA-8, synthesized with an oligomerization time of 6 hours, yielded solid products, indicating successful progression of the reaction. The shorter oligomerization time likely limited the formation of linear oligomers with sufficient chain length and reactivity for the subsequent depolymerization and ring-opening reactions. As a result, the oligomers generated were insufficient or not adequately structured for efficient ring-opening polymerization, leading to the no formation of PDLLA. This suggests that the oligomerization time variable plays a critical role in achieving suitable intermediates for the synthesis of high-molecular-weight PDLLA.
This section now provides a clearer mechanistic rationale for the absence of polymer formation observed in these codes.
4. Comment:
The authors claimed in Figure 7 that the lower the catalyst concentration, the higher the average molecular weight. However, in Table 3, the molecular weight of PLA-7 is presented as lower than that of PLA-8. The two values should be changed. Also, please double-check the points pointed out by the reviewer throughout the text.
Response:
[Line 481] Thank you for identifying this inconsistency. The confusion originated from a rounding issue in the decimal places of the molecular weight values. This has now been corrected in Table 3, and we have performed a thorough check of all related data throughout the manuscript to ensure accuracy and consistency.
5. Comment:
The authors concluded from the analysis of Figure 7 that the longer the oligomerization time, the longer the ROP time, and the lower the catalyst concentration, the higher the average molecular weight. However, it is difficult to completely prove the authors' claim with only two points of experiment. The experimental results for the intermediate conditions of oligomerization time = 5 hours, ROP time = 5 hours, and catalyst concentration = 1.5% for the three factors should be added to Figure 7.
Response:
We fully understand the Reviewer’s concern and agree that a broader set of experimental points would strengthen the statistical analysis. However, the present study was not intended to optimize or determine ideal parameters, but rather to identify key processing variables that influence the molecular weight of PDLLA under our experimental conditions. For this reason, we designed a 2³ full factorial design, which allows us to evaluate the effects and interactions of the factors under study at two extreme levels.
We recognize that the use of words such as "optimize" or "maximize" may have been inappropriate in this context. Accordingly, we have revised the text throughout the manuscript to reflect that our aim was to explore and identify critical variables rather than define optimal conditions. We believe that more comprehensive optimization would require a different experimental strategy, such as response surface methodology (RSM), which is beyond the scope of this initial exploration.
We appreciate the Reviewer’s suggestion and agree that future work could include additional intermediate conditions to further validate the trends observed.
We again thank the Reviewer for these thoughtful comments, which have helped us strengthen the clarity and scientific rigor of our manuscript.
Sincerely,
The Authors
Reviewer 3 Report
Comments and Suggestions for Authors
The manuscript describes an interesting study in which poly-DL-lactic acid was synthesized, characterized, and 3D-printed to demonstrate the potential use as a scaffold for biomedical applications. The work is both relevant and interesting for the scientific community. However, the manuscript needs to be improved in order to be considered for publication.
- In section 1. Introduction, the aim and relevance of the study should be clearly addressed.
- In section 2, it is claimed that a full experimental design was conducted to determine the conditions under which the polymer with the highest molecular weight was obtained. The molecular weight was estimated from experimental viscosity measurements using parameters K=0.0066 and alpha=0.67. More detail is needed to understand the reason why those values for K and alpha are suitable for this polymer.
- Section 2.3 describes that during the synthesis of poly(DL-lactide) the working pressure was 15 microns, which is confusing. Is it micron of mercury?
- Also in section 2.3, it is described that the temperature was controlled with precision of +/- 3. The units of this magnitude should be stated.
- In section 3.3 the FTIR characterization of PDLLA is presented. More information is needed to conclude that the spectrum of the obtained samples corresponds to PDLLA. For instance, a comparison of the characteristic peaks of standard PDLLA and the peaks obtained in the characterization of the samples.
- In page 12, line 536 it is stated that Table 5 presents an ANOVA analysis, but Table 5 is actually a description of the PDLLA degradation temperatures. The information of the ANOVA analysis is very relevant to the study and it is not shown in the manuscript.
- Figure 8 shows the Pareto chart of standardized effects for molecular weight. The meaning and relevance of the dotted line with a value of 2.11 should be discussed.
- In this study a two level experimental design was conducted. This particular design is useful to explore and identify the relevant factors affecting a response (in this case the response was molecular weight). In order to determine the optimal level of the relevant factors, it is necessary to conduct a three level experimental design. Thus, the authors should incorporate a 3 level experimental design to this work, so that the aim of determining the optimal conditions to maximize the molecular weight of the polymer can be fulfilled.
Author Response
We thank the Reviewer for the careful reading and constructive suggestions. All comments have been addressed in the revised manuscript, as detailed below:
1. Comment:
In section 1. Introduction, the aim and relevance of the study should be clearly addressed.
Response:
[Section 1] We appreciate the suggestion. The Introduction has been revised to more clearly define both the objective and the relevance of the study. The objective is explicitly stated in lines 51–56, where we explain that the purpose of this research is to synthesize high-molecular-weight poly-DL-lactic acid (PDLLA) via ring-opening polymerization (ROP) and to explore the influence of key reaction parameters—namely oligomerization conditions, reaction times, and catalyst concentration—in order to fabricate and characterize 3D-printed PDLLA scaffolds. These scaffolds were evaluated in terms of their morphological and mechanical properties and compared to human trabecular bone.
On the other hand, the relevance of this study—particularly its contribution to the development of biodegradable scaffolds for bone tissue engineering—has been emphasized in the final paragraph of the Introduction.
2. Comment:
In section 2, it is claimed that a full experimental design was conducted to determine the conditions under which the polymer with the highest molecular weight was obtained. The molecular weight was estimated from experimental viscosity measurements using parameters K = 0.0066 and α = 0.67. More detail is needed to understand the reason why those values for K and α are suitable for this polymer.
Response:
[Lines 139-145] The manuscript has been updated to include the necessary information. The constants K = 0.0066 and α = 0.67 were selected based on previously published studies involving polylactic acid in chloroform at 25 °C, under conditions comparable to those used in this work.
3. Comment:
Section 2.3 describes that during the synthesis of poly(DL-lactide), the working pressure was 15 microns, which is confusing. Is it micron of mercury?
Response:
[Line 82] Thank you for pointing out this ambiguity. The manuscript now specifies that the working pressure was 15 μmHg (microns of mercury).
4. Comment:
Also in section 2.3, it is described that the temperature was controlled with precision of ±3. The units of this magnitude should be stated.
Response:
[Line 84] The missing units have been added. The temperature was controlled with a precision of ±3 °C.
5. Comment:
In section 3.3, the FTIR characterization of PDLLA is presented. More information is needed to conclude that the spectrum of the obtained samples corresponds to PDLLA. For instance, a comparison of the characteristic peaks of standard PDLLA and the peaks obtained in the characterization of the samples.
Response:
[Page 10] This observation has been addressed. Figure 4 in the revised manuscript shows the FTIR spectrum of a standard PDLLA sample from the OPUS software database, which was used for comparison. The matching peaks are discussed to validate the identity of the synthesized polymer.
6. Comment:
In page 12, line 536, it is stated that Table 5 presents an ANOVA analysis, but Table 5 is actually a description of the PDLLA degradation temperatures. The information of the ANOVA analysis is very relevant to the study and it is not shown in the manuscript.
Response:
[Line 704] We thank the reviewer for highlighting this oversight. The ANOVA table was mistakenly omitted from the original submission. This table has now been correctly included in the manuscript. Please note that this table now appears as Table 6.
7. Comment:
Figure 8 shows the Pareto chart of standardized effects for molecular weight. The meaning and relevance of the dotted line with a value of 2.11 should be discussed.
Response:
[Lines 728-741] The revised manuscript now includes a discussion of the dotted line. This threshold corresponds to the t-value for statistical significance (p=0.05), allowing for identification of the most influential factors in the factorial design.
8. Comment:
In this study a two-level experimental design was conducted. This particular design is useful to explore and identify the relevant factors affecting a response (in this case the response was molecular weight). In order to determine the optimal level of the relevant factors, it is necessary to conduct a three-level experimental design. Thus, the authors should incorporate a 3-level experimental design to this work, so that the aim of determining the optimal conditions to maximize the molecular weight of the polymer can be fulfilled.
Response:
We fully agree with this observation. We acknowledge that the use of terms such as "optimize" or "maximize" in the original version of the manuscript was inappropriate, as they suggest a level of experimental resolution that this study does not aim to achieve. The purpose of our research was to explore key reaction parameters under our working conditions that influence the formation of high-molecular-weight PDLLA. As such, we have revised the manuscript to replace terms implying optimization with language that more accurately reflects the exploratory nature of the study.
Additionally, we have clarified that the choice of a two-level factorial design was intentional and suitable for identifying the most significant factors. A three-level design, as correctly indicated by the reviewer, would be necessary for determining optimal conditions. We appreciate the reviewer’s comment and have incorporated this clarification in the revised text. We also sincerely regret the earlier imprecision in terminology and have corrected it accordingly.
We again thank the Reviewer for these thoughtful comments, which have helped us strengthen the clarity and scientific rigor of our manuscript.
Sincerely,
The Authors
Round 2
Reviewer 2 Report
Comments and Suggestions for Authors
None
Author Response
We sincerely thank you for your valuable suggestions and comments provided throughout the review process. Your feedback has contributed meaningfully to the improvement of our manuscript, and we greatly appreciate the time and dedication invested in evaluating our work.
We would also like to express our commitment to taking your observations into consideration for future research we may submit, as we fully recognize the importance of clarity and thoughtful scientific communication for the effective dissemination of research outcomes.
Thank you once again for your contribution to the enhancement of our study.
Reviewer 3 Report
Comments and Suggestions for Authors
The authors have substantially addressed all my concerns. In my opinion, the manuscript could be accepted for publication.
Author Response

(The authors gave the same response as above.)
